# Severity and Bothersomeness of Urinary Tract Infection Symptoms in Women before and after Menopause

**DOI:** 10.3390/antibiotics12071148

**Published:** 2023-07-04

**Authors:** Signe Teglbrænder-Bjergkvist, Volkert Siersma, Anne Holm

**Affiliations:** The Research Unit for General Practice and Section of General Practice, University of Copenhagen, 1165 Copenhagen, Denmark; knm570@alumni.ku.dk (S.T.-B.); tjs657@sund.ku.dk (V.S.)

**Keywords:** urinary tract infections, symptoms, general practice, menopause, patient-reported outcome measures

## Abstract

Urinary tract infection (UTI) is a common cause for prescription of antibiotics among women in general practice. Diagnosis is often established by inquiry into clinical history and symptoms, and these may be experienced differently depending on menopause status of the woman. The aim of this study was to assess differences in severity and bothersomeness of UTI symptoms between pre- and postmenopausal women. We used a convenience sample of 313 women with suspected UTIs and typical symptoms recruited in general practice. Each woman completed the Holm and Cordoba UTI score (HCUTI), measuring the severity and bothersomeness of the dimensions: dysuria, frequency, lower back, and general symptoms. The exposure was menopausal status. Differences in the various HCUTI dimensions between the menopause groups were investigated in linear regression models, adjusting for potential confounders. Premenopausal women had a significantly higher severity score for the item “feeling unwell” than postmenopausal women (mean difference −0.59, 95% CI −0.88 to −0.31). They also had a significantly higher bothersomeness score for the items “pain on urination” (mean difference −0.54, 95% CI −0.83 to −0.25), “feeling unwell” (mean difference −0.62, 95% CI −0.92 to −0.32), and for the dimension “dysuria” (mean difference −0.38, 95% CI −0.61 to −0.15) than postmenopausal women. This study found differences in some aspects of symptom severity and bothersomeness between pre- and postmenopausal women presenting in general practice with suspected UTIs. Menopausal status should be taken into account when using symptoms to diagnose and evaluate response to UTI treatment in both clinical practice and research.

## 1. Introduction

Urinary tract infection (UTI) is a common cause for prescription of antibiotics in general practice, the second-most common after respiratory tract infections [1,2]. Antibiotic use in primary care for UTIs is associated with antimicrobial resistance [3,4]. Since patients mostly benefit from antibiotics if they have significant bacteriuria, it is important to establish an accurate diagnosis before initiating treatment [5]. Although a lower UTI is mostly harmless, the symptoms can be distressing and affect quality of life considerably, leading to most women receiving a prescription of antibiotics when a UTI is suspected [6,7,8]. It is recommended to use urine tests for diagnosing UTIs, but the diagnosis is often established based on the clinical history and reported symptoms [9,10]. Women who have experienced having a UTI report a number of diverse symptoms, among others: burning pain when urinating, pain around the bladder, uncomfortable pressure in the lower abdomen and back, frequent voiding, incontinence, and generally feeling unwell [11,12,13,14]. A few studies have indicated that premenopausal women have a higher symptom severity for some symptoms than postmenopausal women [15,16]. In addition, a recent study suggested that the age of the patient may affect the prevalence and diagnostic value of the symptoms recorded in general practice [17]. However, it is unclear if the difference between pre- and postmenopausal women was due to a different symptom experience or a different reporting or recording of symptoms in the practice. Since the symptoms are used to establish the diagnosis, it is of importance to investigate whether menopausal status affects which symptoms the women experience. If the symptom-experience when having a UTI differs with menopausal status, this may affect the diagnostic value of symptoms and signs in clinical practice, as well as the relevance of individual symptoms. Further, this difference in the diagnostic value of symptoms may lead to some women being misdiagnosed and either under- or overtreated with antibiotics [18].

The aim of this study was to assess differences in severity and bothersomeness of UTI symptoms between pre- and postmenopausal women using the Holm and Cordoba UTI score (HCUTI), a validated patient-reported outcome measure (PROM).

## 2. Results

During the recruitment period, 20 practices chose to participate in the study and included 313 women (see Figure 1).

Table 1 presents the distribution of baseline variables. A total of 66.7% of the women had a significant bacteriuria. Premenopausal women had significant bacteriuria in 56% of cases, while postmenopausal women had significant bacteriuria in 63% of cases. The most frequent pathogen-variable in all age groups was “Primary pathogen”. Patients usually waited three days before consulting a doctor in all age groups. Postmenopausal women had taken painkillers within the last 24 h in 41% of cases, while the premenopausal women had taken painkillers in 24% of cases. All co-variates except for days with symptoms before consultation were unevenly distributed between the groups. A total of 153 patients had no missing data, and missing values were randomly distributed throughout the dataset. Symptom severity scores for the HCUTI are presented in Table 2. The first three columns present the raw symptom scores for each of the dimensions dysuria, frequency, lower back, and general symptoms, as well as the individual symptoms included in each dimension and six individual symptoms. The rightmost column presents the adjusted difference between the symptom score of the premenopausal women and the postmenopausal women. Premenopausal women had significantly higher adjusted symptom severity scores than postmenopausal women in the dimensions dysuria (adjusted difference; −0.30 95% CI; (−0.51 to −0.08)) and general symptoms (adjusted difference; −0.26, 95% CI; (−0.45 to −0.06)), as well as the individual symptoms “pain on urination”, “uncomfortable pressure around the bladder”, “feeling unwell”, “blood in the urine”, and “pain around the bladder”. Postmenopausal women had a significantly higher adjusted symptom severity score on the symptom “incontinence”. However, after adjusting for multiple testing, only one symptom score, “feeling unwell”, remained significantly higher in premenopausal women (adjusted difference; −0.59, 95% CI: (−0.88 to −0.31)).

Symptom bothersomeness scores are presented in Table 3, similar to Table 2. Premenopausal women had significantly higher symptom bothersomeness scores than postmenopausal women in the dimensions dysuria (adjusted difference; −0.38, 95% CI: (−0.61 to −0.15)) and general symptoms (adjusted difference; −0.26, 95% CI: (−0.46 to −0.06)), as well as the individual symptoms “pain on urination”, “uncomfortable pressure around the bladder”, “feeling unwell”, and “pain around the bladder”. For the bothersomeness construct, the symptoms “Pain on urination” (adjusted difference; −0.54, 95% CI: (−0.83 to −0.25)), “Feeling unwell” (adjusted difference; −0.62, 95% CI: (−0.92 to −0.32)), and the dimension “Dysuria” remained significantly higher in premenopausal women after adjusting for multiple testing.

## 3. Discussion

In this study with 313 women with suspected UTI, we found that premenopausal women had a significantly higher severity score for the item “feeling unwell” than postmenopausal women. They also had a significantly higher bothersomeness score for the items “pain on urination” and “feeling unwell” and the dimension “dysuria” than postmenopausal women.

The study has a number of strengths. Our outcome was the validated HCUTI PROM, making it likely that we captured actual patient experiences. Although we had missing data, our statistical method made it possible to analyse all patients, making our results more robust. We also had a sufficient number of patients in all groups to capture significant differences, even after adjusting for multiple testing.

The study also has its limitations. We did not collect any information about the participants’ menstrual cycles. Thus, we cannot be sure if the participants were actually physiologically pre-, peri-, or postmenopausal. There is a risk of sample selection bias: the GPs were recruiting for a randomized trial in which patients had to wait one day before they could start on antibiotics. This could have made the GPs more restrictive when including older patients in the study. Thus, more frail elderly are not likely to be represented in our study. There was considerable missing data in various variables, but bias and loss of power was avoided by the use of multiple imputation. Since this is a post hoc analysis and it includes multiple analyses, there is a risk of type 1 errors. We have addressed this problem by adjusting for multiple testing by the method of Benjamini-Hochberg. This was an explorative study, and we did not perform a power calculation, but with the small sample size included, there is a risk of type 2 errors. Hence, we cannot claim that we have found all differences, but we can be confident about the differences that were found [19].

Previous studies have found that postmenopausal women more often complained about “Lower abdominal pain” than premenopausal women [16]. This is in contrast with our findings, where we observed higher scores in a similar symptom (“Pain around the bladder”) in premenopausal women compared with postmenopausal women. They also reported that premenopausal women more often complained about the symptoms “Painful urination” and “Burning urination” than postmenopausal women. We observed higher scores in “Painful urination” in premenopausal women, but we found no significant difference between groups in “Burning urination”. Finally, they examined a symptom they called “General malaise”, which corresponds to our symptom “Feeling unwell”, and found no difference between groups. In contrast, we found higher scores of “Feeling unwell” in the premenopausal group. In a study from 2003 on 398 women presenting in general practice, they found that the symptom “Pressure in the genital area” was more frequently present in younger women and the symptom “Low backache” had a skewed distribution in frequency, with the highest values in the age group 51–65 years [15]. They examined how many of the patients across different age groups had the presence of symptoms. They did not find any difference in symptoms related to pain on urination. In regard to general symptoms such as “Unwell” and “Weak, tired or not in good form”, they also found skewed distributions, with the highest values in the age group 51–65 years, although these differences were not significant. Thus, findings in different studies are not quite similar, although they all find differences. This could be due to the use of different outcome measures and different definitions of menopausal status.

The trend in our results with higher severity of pain, feeling unwell, and blood in the urine could point in a direction where premenopausal women have a more severe infection with a higher level of tissue damage or a more powerful immune response [20,21]. However, there could be numerous additional explanations for these findings, and our results remain hypothesis-generating and should be repeated by other research teams.

## 4. Materials and Methods

### 4.1. Inclusion Criteria

This study is a post hoc analysis of previously collected data from women presenting in general practice with suspected UTIs. The data were collected in connection with a diagnostic randomized controlled trial assessing two diagnostic tests for UTIs [22]. A more thorough description of the methods can be seen in the protocol for the diagnostic trial [23]. A random selection of 200 general practitioners (GPs) in the Copenhagen area was invited. Recruitment took place between 1 March 2015 and 1 May 2016. Participants were identified at their GP when presenting with at least one symptom of a UTI (dysuria, frequency or urge) lasting for 7 days or less and meeting the inclusion criteria: female, >18 years, unpregnant, no urological abnormalities, immunocompetent, able to deliver a mid-stream urine sample, and able to provide informed consent as judged by the general practitioner.

### 4.2. Data Collection

Data in this study were collected from three sources: a case report form, which the GP or practice staff completed, a symptom diary that the participants completed, and the microbiological urine culture results collected from the database at the microbiological laboratory. After the patient had signed informed consent, the GP completed the case report form on the day of the consultation, including name and social security number, number of UTIs within the past year, and duration of symptoms before the consultation. Urine samples were sent to either Department of Clinical Microbiology, Copenhagen University, Hospital, Herlev, Denmark or the Department of Clinical Microbiology, Copenhagen University Hospital, Hvidovre, Denmark. Urine samples were analyzed on Inoqul A Bi-plate (CHROMagar and blood agar) with 10 μL on each half of the agar. All samples were quantified. Significant growth was defined as growth of ≥10^3^ cfu/mL for *Escherichia coli* and *Staphylococcus saprophyticus*, ≥10^4^ cfu/mL for other typical uropathogens, and ≥10^5^ cfu/mL for possible uropathogens in line with the European guidelines for urinalysis [24]. Plates with growth of more than two uropathogens were labelled as mixed cultures and classified in the analysis as negative. The HCUTI was handed out after inclusion and completed at nighttime on the same day. 

### 4.3. Variables

The exposure was menopausal status defined, in lack of direct information, as: (1) Premenopausal (<45 years), (2) perimenopausal (≥45 and <60 years), and (3) postmenopausal (≥60 years). The analysis compared the premenopausal group with the postmenopausal group. The outcome was measured with the HCUTI [25]. The HCUTI scores the severity and bothersomeness, respectively, of 18 symptoms on a 4-point Likert scale (0 to 3). Some of the symptoms were gathered into four dimensions for both the severity and bothersomeness scores: dysuria (3 symptoms), frequency (4 symptoms), lower back (2 symptoms), and general (3 symptoms), and the remaining 6 symptoms are included in the HCUTI outcome portfolio individually. The Danish version of the HCUTI score was developed and validated for women presenting with UTI symptoms in primary care. Based on clinical assumptions, we registered the following potential confounders use of painkillers, days of symptoms before consultation, number of previous UTIs within the past year, social class, employment, and pathogen. In this study, we have categorized urine samples with significant growth of *E. Coli* and *S. Saprophyticus* as “Primary uropathogens”, samples with significant growth of one or two other uropathogens as “Other uropathogens”, and samples with no growth or growth of more than 2 uropathogens as “No uropathogens or contaminated”. Duration of symptoms before consultation were categorized for analysis as 1, 2, 3, 4, 5, 6, or 7 days and the number of previous UTIs within the last year as 1, 2, 3 or 4+. Information about the use of pain killers was a binary “Yes/No” category.

### 4.4. Statistical Methods

Differences in the distribution of baseline variables between groups were investigated using the chi-squared test for categorical data and analysis of variance (ANOVA) for continuous variables. Differences in symptom scores between the menopause status groups were investigated in linear regression models, both an unadjusted model and adjusted for possible confounders. The method of multiple imputation was used to avoid bias and loss of power because of missing values for various covariates. This approach constructs five sets of data where the missing values are imputed with (stochastic) guesses using chained equations, and the analyses results from these five data sets are thereafter combined using Rubin’s rule [26]. Further, to adjust for multiple testing, we controlled the false discovery rate at 5% by the method of Benjamini-Hochberg [27]. All analyses were performed in SAS v 9.

## 5. Conclusions

Premenopausal women generally experienced symptoms related to pain on urination and feeling unwell as more severe and bothersome than postmenopausal women did. The newly developed core outcome set for the evaluation of trials regarding UTIs recommends several outcomes related to symptom resolutions [28]. The differences in symptom scores in this study were around 0.5 points, corresponding to half of women moving from “a little” to “some” or “some” to “a lot”. This could significantly affect the time to resolution in trials. We would generally consider this a clinically significant difference. This study indicates that menopause status should be taken into account when using symptoms to diagnose and evaluate response to UTI treatment, both in clinical practice and research.

## Figures and Tables

**Figure 1 antibiotics-12-01148-f001:**
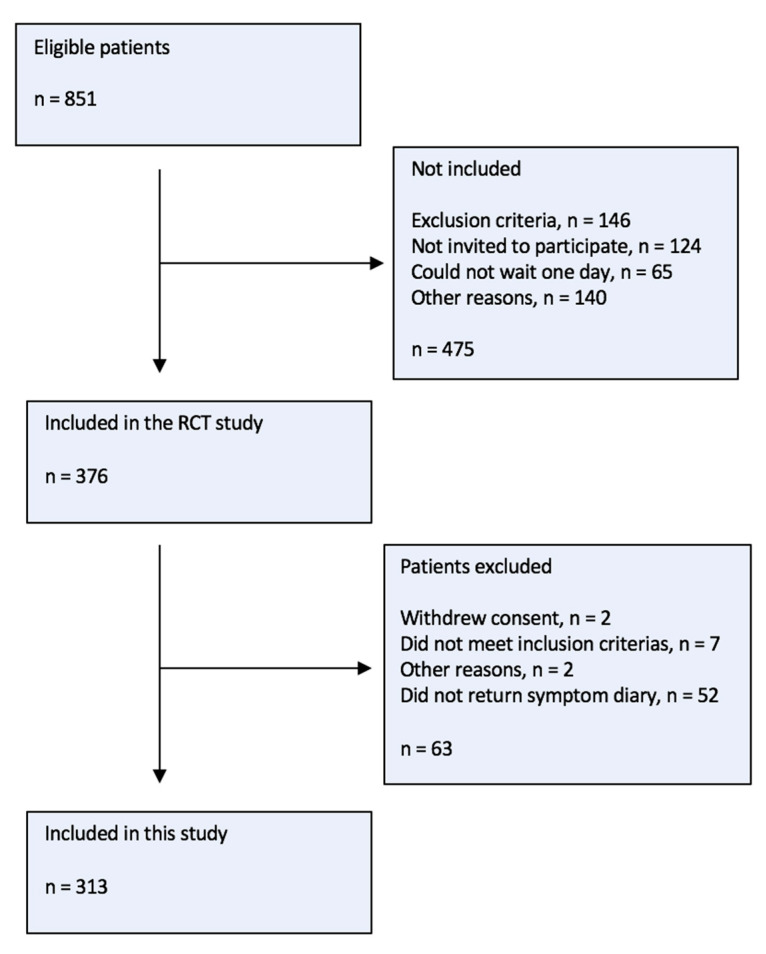
Inclusion flow chart.

**Table 1 antibiotics-12-01148-t001:** Study participant characteristics. Participants are divided in three age groups: premenopausal < 45; perimenopausal 45–59; postmenopausal > 60.

	Premenopausal (*n* = 125)	Perimenopausal (*n* = 76)	Postmenopausal (*n* = 110)	*p*
Pathogen (*n* missing = 18), *n* (%)				0.0023
No pathogen	52 (44%)	15 (20%)	38 (37%)	
Primary pathogen	60 (51%)	52 (68%)	52 (50%)	
Other uropathogen(s)	5 (4%)	7 (9%)	14 (13%)	
Use of painkillers (*n* missing = 16) *n* (%)	28 (24%)	23 (31%)	44 (41%)	0.0245
Days of symptoms before consultation (*n* missing = 11), median [QR1;QR3]	3 [2;5]	3 [1;4]	3 [2;6]	0.1043
Number of previous UTI within past year (*n* missing = 13), median [QR1;QR3]	0 [0;1]	1 [0;2]	1 [0;2]	0.0422
Social class (*n* missing = 122), *n* (%)				0.0002
1 (highest)	5 (7%)	7 (14%)	3 (4%)	
2	24 (33%)	15 (29%)	24 (36%)	
3	3 (4%)	3 (6%)	8 (12%)	
4	14 (19%)	15 (29%)	29 (43%)	
5 (lowest)	27 (37%)	11 (22%)	3 (4%)	
Employment (*n* missing = 13), *n* (%)				<0.001
Working	68 (59%)	64 (85%)	21 (19%)	
During education	37 (32%)	1 (1%)	0	
Job seeking	6 (5%)	3 (4%)	0	
Early retirement pay, stay-at-home, etc.	5 (4%)	7 (9%)	88 (81%)	

**Table 2 antibiotics-12-01148-t002:** Symptom severity score outcome by groups of menopause status (pre-, peri-, and postmenopausal women). Shown are means with 95% confidence intervals and differences with 95% confidence intervals calculated by an adjusted linear regression model using the symptom severity scores as the outcome and groups of menopause status (pre-, peri-, and postmenopausal) as the exposure. Co-variates in the adjusted model were; the number of previous urinary tract infections, days with symptoms before consulting, bacteriuria, and use of painkillers. * = significant on a 5% significance level, ** = significant after adjusting with the Benjamini-Hochberg method. The method of Benjamini-Hochberg rejects all *p*-values less than 0.0028 to control the false discovery rate at 5%.

Symptom	Premenopausal	Perimenopausal	Postmenopausal	Adjusted Difference between Postmenopausal and Premenopasal Women
DYSURIA Symptom dimension	1.54 (1.48 to 1.61)	1.47 (1.39 to 1.55)	1.31 (1.24 to 1.38)	−0.30 (−0.51 to −0.08) *
Pain on urination	1.76 (1.68 to 1.85)	1.89 (1.79 to 1.99)	1.40 (1.31 to 1.50)	−0.41 (−0.68 to −0.13) *
Difficulty to empty bladder	1.32 (1.24 to 1.41)	1.29 (1.18 to 1.41)	1.27 (1.18 to 1.37)	−0.14 (−0.45 to 0.16)
Uncomfortable pressure around the bladder	1.55 (1.46 to 1.64)	1.24 (1.13 to 1.35)	1.26 (1.17 to 1.35)	−0.34 (−0.63 to −0.04) *
FREQUENCY Symptom dimension	1.69 (1.62 to 1.77)	1.77 (1.69 to 1.86)	1.85 (1.78 to 1.92)	0.12 (−0.11 to 0.35)
Frequent urination–daytime	2.09 (2.00 to 2.17)	2.22 (2.12 to 2.32)	2.06 (1.99 to 2.14)	−0.08 (−0.34 to 0.19)
Increased urge for urination	2.15 (2.07 to 2.23)	2.29 (2.20 to 2.39)	2.12 (2.04 to 2.20)	−0.07 (−0.31 to 0.18)
Has to hurry to the toilet	1.57 (1.48 to 1.67)	1.59 (1.47 to 1.71)	1.80 (1.70 to 1.90)	0.21 (−0.10 to 0.51)
Incontinence	0.97 (0.88 to 1.06)	1 (0.89 to 1.11)	1.42 (1.32 to 1.51)	0.41 (0.11 to 0.71) *
LOWER BACK Symptom dimension	0.54 (0.47 to 0.61)	0.60 (0.51 to 0.68)	0.54 (0.46 to 0.61)	−0.06 (−0.29 to 0.17)
Pain in lower back	0.61 (0.54 to 0.68)	0.64 (0.55 to 0.74)	0.62 (0.54 to 0.70)	−0.05 (−0.3 to 0.20)
Uncomfortable pressure in lower back	0.47 (0.40 to 0.54)	0.55 (0.46 to 0.64)	0.46 (0.38 to 0.53)	−0.08 (−0.31 to 0.16)
GENERAL Symptom dimension	0.85 (0.80 to 0.90)	0.88 (0.79 to 0.97)	0.66 (0.60 to 0.72)	−0.26 (−0.45 to −0.06) *
Feeling unwell	1.58 (1.49 to 1.66)	1.35 (1.25 to 1.46)	1.10 (1.00 to 1.19)	−0.59 (−0.88 to −0.31) *, **
Fever	0.48 (0.42 to 0.54)	0.66 (0.56 to 0.76)	0.46 (0.39 to 0.52)	−0.06 (−0.28 to 0.16)
Shivering	0.50 (0.43 to 0.56)	0.63 (0.53 to 0.72)	0.41 (0.34 to 0.48)	−0.12 (−0.35 to 0.11)
Single symptoms				
Burning	1.67 (1.58 to 1.76)	1.67 (1.56 to 1.78)	1.56 (1.47 to 1.66)	−0.20 (−0.50 to 0.11)
Smell	1.24 (1.15 to 1.33)	1.22 (1.10 to 1.34)	1.09 (0.99 to 1.19)	−0.22 (−0.56 to 0.11)
Appearance	1.01 (0.93 to 1.08)	1.19 (1.08 to 1.31)	1.09 (1.01 to 1.18)	0.04 (−0.26 to 0.34)
Blood	0.45 (0.38 to 0.51)	0.53 (0.44 to 0.62)	0.18 (0.13 to 0.23)	−0.28 (−0.53 to −0.03) *
Frequent urination–nighttime	1.22 (1.13 to 1.31)	1.25 (1.14 to 1.37)	1.55 (1.45 to 1.64)	0.27 (−0.04 to 0.58)
Pain around bladder	1.53 (1.44 to 1.61)	1.22 (1.12 to 1.33)	1.2 (1.11 to 1.28)	−0.40 (−0.68 to −0.12) *

**Table 3 antibiotics-12-01148-t003:** Bothersomeness score outcome by groups of menopause status (pre-, peri-, and postmenopausal women). Shown are means with 95% confidence intervals and differences with 95% confidence intervals calculated by an adjusted linear regression model using the bothersomeness scores as the outcome and groups of menopause status (pre-, peri-, and postmenopausal) as the exposure. Co-variates in the adjusted model were; the number of previous urinary tract infections, days with symptoms before consulting, bacteriuria, and use of painkillers. Significant estimates are marked with an asterisk, * = significant on a 5% significance level, ** = significant after adjusting with the Benjamini-Hochberg method. The method of Benjamini-Hochberg rejects all *p*-values less than 0.0028 to control the false discovery rate at 5%.

Symptom	Premenopausal	Perimenopausal	Postmenopausal	Adjusted Difference between Postmenopausal and Premenopasal Women
DYSURIA Symptom dimension	1.64 (1.57 to 1.71)	1.46 (1.38 to 1.55)	1.32 (1.25 to 1.39)	−0.38 (−0.61 to −0.15) *, **
Pain on urination	1.91 (1.81 to 2.00)	1.94 (1.84 to 2.05)	1.41 (1.31 to 1.51)	−0.54 (−0.83 to −0.25) *, **
Difficult to empty bladder	1.48 (1.38 to 1.58)	1.23 (1.11 to 1.35)	1.29 (1.19 to 1.39)	−0.28 (−0.61 to 0.04)
Uncomfortable pressure around the bladder	1.53 (1.43 to 1.62)	1.22 (1.11 to 1.34)	1.26 (1.17 to 1.35)	−0.32 (−0.62 to −0.01) *
FREQUENCY Symptom dimension	1.68 (1.60 to 1.75)	1.64 (1.54 to 1.74)	1.81 (1.73 to 1.89)	0.09 (−0.16 to 0.34)
Frequent urination–daytime	1.95 (1.87 to 2.04)	1.84 (1.72 to 1.96)	1.94 (1.85 to 2.04)	−0.07 (−0.36 to 0.23)
Increased urge for urination	2.05 (1.97 to 2.14)	2.06 (1.95 to 2.17)	2.06 (1.98 to 2.15)	−0.03 (−0.31 to 0.25)
Has to hurry to the toilet	1.59 (1.49 to 1.69)	1.54 (1.42 to 1.66)	1.79 (1.69 to 1.89)	0.17 (−0.14 to 0.49)
Incontinence	1.12 (1.02 to 1.22)	1.13 (1.00 to 1.25)	1.46 (1.35 to 1.56)	0.29 (−0.04 to 0.62)
LOWER BACK Symptom dimension	0.54 (0.47 to 0.61)	0.59 (0.50 to 0.67)	0.51 (0.43 to 0.58)	−0.10 (−0.33 to 0.14)
Pain in lower back	0.60 (0.53 to 0.67)	0.63 (0.54 to 0.72)	0.58 (0.50 to 0.67)	−0.07 (−0.32 to 0.17)
Uncomfortable pressure in lower back	0.48 (0.41 to 0.55)	0.54 (0.45 to 0.63)	0.43 (0.36 to 0.50)	−0.12 (−0.36 to 0.12)
GENERAL Symptom dimension	0.84 (0.78 to 0.89)	0.90 (0.81 to 0.99)	0.65 (0.59 to 0.71)	−0.26 (−0.46 to −0.06) *
Feeling unwell	1.59 (1.50 to 1.69)	1.46 (1.34 to 1.58)	1.11 (1.01 to 1.21)	−0.62 (−0.92 to −0.32) *, **
Fever	0.45 (0.38 to 0.51)	0.65 (0.55 to 0.75)	0.44 (0.37 to 0.51)	−0.05 (−0.28 to 0.17)
Shivering	0.47 (0.40 to 0.54)	0.60 (0.50 to 0.70)	0.4 (0.33 to 0.47)	−0.11 (−0.35 to 0.13)
Single symptoms				
Burning	1.74 (1.64 to 1.83)	1.72 (1.61 to 1.84)	1.56 (1.47 to 1.66)	−0.26 (−0.57 to 0.04)
Smell	0.97 (0.88 to 1.07)	1.02 (0.90 to 1.14)	0.93 (0.83 to 1.03)	−0.08 (−0.41 to 0.25)
Appearance	0.66 (0.59 to 0.73)	0.88 (0.77 to 0.98)	0.88 (0.79 to 0.97)	0.18 (−0.14 to 0.51)
Blood	0.38 (0.31 to 0.45)	0.50 (0.40 to 0.59)	0.19 (0.13 to 0.24)	−0.19 (−0.47 to 0.09)
Frequent urination–nighttime	1.31 (1.21 to 1.40)	1.30 (1.18 to 1.43)	1.57 (1.47 to 1.68)	0.18 (−0.16 to 0.51)
Pain around bladder	1.55 (1.46 to 1.64)	1.22 (1.11 to 1.33)	1.20 (1.11 to 1.29)	−0.41 (−0.71 to −0.11) *

## Data Availability

All authors had access to and can take responsibility for data and analysis. The authors commit to making the relevant anonymised patient level data available on reasonable request.

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
