# Peer review of "Severity and Bothersomeness of Urinary Tract Infection Symptoms in Women before and after Menopause"

_antibiotics, 2023, doi:10.3390/antibiotics12071148_

Round 1

Reviewer 1 Report

State the problem existing in the study population in a clear way- ie., the burden of disease

The rationale of the study has to be improved

Details of the study design have to be mentioned

Statistical analysis can be improved by including logistic regression tests.

The sampling technique is unclear.. Why do the authors rely on nonprobability sampling

The study period is seven years older. The latest data are recommended.

How the authors confirmed the participants are not pregnant and immunocompetent

What is called well-controlled diabetes and elderly women participants

Why the authors set different cutoff values for E. coli and S. saprophyticus has to be denoted

The spelling of S. saprophyticus has to be checked

Follow the binomial system of writing bacterial name

Have the authors performed the serotyping of E.coli to label it as a uropathogenic E.coli

Why do Authors only include S. saprophyticus and E.coli as primary pathogens

What about the antimicrobial susceptibility test results

Implications of the study have to be specified

AVERAGE, CAN BE POLISHED

Author Response

Thank you for your comments.

State the problem existing in the study population in a clear way- ie., the burden of disease

The introduction has been elaborated, please see the revised manuscript

The rationale of the study has to be improved

The introduction has been elaborated, please see the revised manuscript

Details of the study design have to be mentioned

I am not sure which details the reviewer refers to here. I would be happy to provide more details in specific sections.

Statistical analysis can be improved by including logistic regression tests.

In this case, we chose linear regression models since we did not investigate binary outcomes, for which logistic regression models are suited.

The sampling technique is unclear.. Why do the authors rely on nonprobability sampling

We did not use non-probability sampling. We had in- and exclusion criteria.

The study period is seven years older. The latest data are recommended.

This is a post hoc analysis and we do not have newer data. We have put this under the limitations section

How the authors confirmed the participants are not pregnant and immunocompetent

This was confirmed by the GP. We have added this to the methods.

What is called well-controlled diabetes and elderly women participants

We have deleted this section since it was more confusing than informative.

Why the authors set different cutoff values for E. coli and S. saprophyticus has to be denoted

This is in line with the European guidelines for urinalysis, which is used in Denmark [1].

The spelling of S. saprophyticus has to be checked

We have found and corrected the misspell

Follow the binomial system of writing bacterial name

We have corrected the bacterial names throughout the manuscript

Have the authors performed the serotyping of E.coli to label it as a uropathogenic E.coli

No, this is not standard procedure in Denmark. We have performed the analysis stated in the methods.

Why do Authors only include S. saprophyticus and E.coli as primary pathogens

This is in line with the European guidelines for urinalysis, which is used in Denmark [1].

What about the antimicrobial susceptibility test results

These were performed in the original study, but are not relevant for this study and are therefore not described in the methods or results. We have previously investigated that they did not influence the symptoms [2].

Implications of the study have to be specified

The clinical implications are stated in the end of the discussions section “This study indicates that age and menopause status should be taken into account when using symptoms to diagnose and evaluate response to UTI treatment both in clinical practice and research.”

  1. Aspevall, O.; Hallander, H.; Gant, V.; Kouri, T. European Guidelines for Urinalysis: A Collaborative Document Produced by European Clinical Microbiologists and Clinical Chemists under ECLM in Collaboration with ESCMID. Scand. J. Clin. Lab. Invest. 2000, 60, 1–96.
  2. Waldorff, M.S.; Bjerrum, L.; Holm, A.; Siersma, V.; Bang, C.; Llor, C.; Cordoba, G. Influence of Antimicrobial Resistance on the Course of Symptoms in Female Patients Treated for Uncomplicated Cystitis Caused by Escherichia Coli. Antibiotics 2022, 11.

Reviewer 2 Report

This is a straightforward study, indicating differences in severity and bothersomeness between women of different menopause status, who addressed a general practice doctor with suspected urinary tract infections (UTIs). The results obtained show that age and menopause status should be taken into account when symptoms are used to diagnose and evaluate response to UTI treatment, which is of large relevance in both clinical practice and research. Using a validated UTI score (Holm and Cordoba), taking missing data into account, and having a sufficient number of patients in all groups provide robustness of the results. The study has some limitations, but these are addressed by the authors. In general, the manuscript is well-structured, methodologically sound, clearly presented, and can be published in its present form.

Author Response

Thank you

Reviewer 3 Report

I have read with interest the manuscript submitted by Teglbrænder-Bjergkvist et al.

I have some comments to be addressed in order to improve the quality of this manuscript:

- the bacterial name should be italicized; the first time a bacterial name is used in the manuscript, it should be written entirely, then use the abbreviated form.

- if comparing pre- versus post-menopausal symptoms, can the authors explain why "elderly" women were excluded? Moreover, the post-monopausal group is defined as ">=60 years", with no superior age limit. Also, why exclude only patients with well-controlled diabetes?

- S. Saprofyticus??

The article is, in my opinion, too short and it contains limited information. The tables are hard to understand (maybe they should be fragmented in order to provide a better understanding) and the results section contains only 2 paragraphs..

Consider adding supplementary information about the patients (such as but not limited to associated pathology, previous and current treatment/hospitalizations, outcome, and laboratory findings). Also, consider adding some figures.

The material and methods section lacks significant information about the microbiological methods utilized.

The refference list should is not written according to the MDPI pattern and it is, as well, brief.

Minor spell checks.

- in the tables some words are capitalized while others are not. Please provide a uniform pattern.

- bacterial names are written incorectly throughout the manuscript.

Author Response

- the bacterial name should be italicized; the first time a bacterial name is used in the manuscript, it should be written entirely, then use the abbreviated form.

We have corrected the bacterial names

- if comparing pre- versus post-menopausal symptoms, can the authors explain why "elderly" women were excluded? Moreover, the post-monopausal group is defined as ">=60 years", with no superior age limit. Also, why exclude only patients with well-controlled diabetes?

We actually did not exclude elderly women, but it was stated somehow confusing. The sentence is now deleted. The post-menopausal group did not have an upper limit. This is correct.

- S. Saprofyticus??

We apologize. It has been corrected

The article is, in my opinion, too short and it contains limited information. The tables are hard to understand (maybe they should be fragmented in order to provide a better understanding) and the results section contains only 2 paragraphs..

The manuscript has become longer after revision, but the study is quite small and do not warrant a long manuscript. We agree, that the tables are quite large. We have deleted the columns concerning the perimenopausal women in table 2 and 3, since we do not refer to them in the manuscript. We have kept them in table 1 for transparency.

Consider adding supplementary information about the patients (such as but not limited to associated pathology, previous and current treatment/hospitalizations, outcome, and laboratory findings). Also, consider adding some figures.

We have added all the available variables in table 1. Some of the variables, you mention are in this table. We do not have he rest.

The material and methods section lacks significant information about the microbiological methods utilized.

We have now referred to the protocol in which all information about the microbiological methods is described. Since this article is not about microbiology, we do not wish for it to take up too much space.

The refference list should is not written according to the MDPI pattern and it is, as well, brief.

The reference list has expanded and corrected to the journals standards.

Round 2

Reviewer 3 Report

The manuscript has become just briefly longer.

If no further information is available for the results section, at least the introduction and discussion sections should be expanded, considering that this topic is well-studied and plenty of information is available.

In addition, given the fact that the tables are long and quite arid, you can take some of the information and transform it into some figures.

Avoid further self-citations. 20% of all citations are self-citations.

"We have now referred to the protocol in which all information about the microbiological methods is described. Since this article is not about microbiology, we do not wish for it to take up too much space." - lack of space is not a problem in this case.

Moderate English editing is required.

Author Response

Dear reviewer. Thank you for your suggestions for improvement. We have gone through the manuscript one more time and hope you will find it improved.

The manuscript has become just briefly longer.

If no further information is available for the results section, at least the introduction and discussion sections should be expanded, considering that this topic is well-studied and plenty of information is available.

The results section and the introduction have been expanded as well as a little in the discussions. We hope, you will find the longer sections an improvement.

In addition, given the fact that the tables are long and quite arid, you can take some of the information and transform it into some figures.

We have taken your previous advice and put some of the results in the text. We prefer tables to figures since these presents the raw numbers, which we find of importance here. But we have deleted a column in table 2 and 3 to try and make them easier to read.

Avoid further self-citations. 20% of all citations are self-citations.

We have not added any. The existing self-capitations refers to 1) the study that started our curiosity regarding this topic (reference 17), 2) the original study from which the data came (references 23 and 24) and the validation study of the PROM used in the trials (reference 26). We do not think any of these can be removed without loosing transparency. 

"We have now referred to the protocol in which all information about the microbiological methods is described. Since this article is not about microbiology, we do not wish for it to take up too much space." - lack of space is not a problem in this case.

No, the article is now 2.200 words, which is not long. We still think the microbiological methods are decribed in a sufficient detail for a co-variate and have not added any additional information.

Round 3

Reviewer 3 Report

I agree, the manuscript has improved, even though the authors refused the proposed changes, which would have strengthened the article both scientifically and visually.

Some spell checks/punctuation are required.